# Factors influencing implementation of an insulin patient decision aid at public health clinics in Malaysia: A qualitative study

**Wen Ting Tong[1], Yew Kong Lee[1]\*, Chirk Jenn Ng[1], Ping Yein Lee[2]**

**1** Department of Primary Care Medicine, Faculty of Medicine, University of Malaya, Kuala Lumpur, Malaysia, **2** Department of Family Medicine, Faculty of Medicine and Health Sciences, University Putra Malaysia, Serdang, Malaysia

\* leeyk@um.edu.my

**Data Availability Statement:** All relevant data are within the paper and its Supporting information files.

## Abstract

### Background

Many patient decision aids (PDAs) are developed in academic settings by academic researchers. Academic settings are different from public health clinics where the focus is on clinical work. Thus, research on implementation in public health settings will provide insights to effective implementation of PDA in real-world settings. This study explores perceived factors influencing implementation of an insulin PDA in five public health clinics.

### Methods

This study adopted a comparative case study design with a qualitative focus to identify similarities and differences of the potential barriers and facilitators to implementing the insulin PDA across different sites. Focus groups and individual interviews were conducted with 28 healthcare providers and 15 patients from five public health clinics under the Ministry of Health in Malaysia. The interviews were transcribed verbatim and analysed using the thematic approach.

### Results

Five themes emerged which were: 1) time constraint; 2) PDA costs; 3) tailoring PDA use to patient profile; 4) patient decisional role; and 5) leadership and staff motivation. Based on the interviews and drawing on observations and interview reflection notes, time constraint emerged as the common prominent factor that cut across all the clinics, however, tailoring PDA use to patient profile; patient decisional role; leadership and staff motivation varied due to the distinct challenges faced by specific clinics. Among clinics from semi-urban areas with more patients from limited education and lower socio-economic status, patients' ability to comprehend the insulin PDA and their tendency to rely on their doctors and family to make health decisions were felt to be a prominent barrier to the insulin PDA implementation. Staff motivation appeared to be stronger in most of the clinics where specific time was allocated to diabetes team to attend to diabetes patients and this was felt could be a potential

**Funding:** This study was supported by the University of Malaya Research Grant (UMRG) (RP041C-15HTM) (YKL, WTT; CJN, PYL); and the University of Malaya Postgraduate Research Grant (PPP) (PG264-2016A) (YKL, WTT, CJN, PYL). The funders had no role in study design, data collection and analysis, decision to publish, or preparation of the manuscript.

**Competing interests:** The authors have declared that no competing interests exist.

facilitator, however, a lack of leadership might affect the insulin PDA implementation even though a diabetes team is present.

## Conclusions

This study found time constraint as a major potential barrier for PDA implementation and effective implementation of the insulin PDA across different public health clinics would depend on leadership and staff motivation and, the need to tailor PDA use to patient profile. To ensure successful implementation, implementers should avoid a 'one size fits all' approach when implementing health innovations.

## Introduction

Patient decision aids (PDAs) are tools to facilitate shared decision making (SDM) between patient and provider. Although shown to be effective in improving decision making [1], their implementation in routine clinical settings is still lacking [2], hampered by various factors such as time constraint, healthcare professionals' attitude, patient characteristics, clinic capacity and processes of care and the healthcare environment [2–4]. Furthermore, there is limited data on PDA implementation in Asia.

Many of the PDAs are implemented at the academic settings [5–7] where the PDAs are usually developed and tested and physicians are researchers and teachers. It is different from public health clinics where the physicians focus on clinical work. Study on implementation in public health setting will be able to provide insights to effective implementation of PDA in the real world settings as new intervention may not work as well as in research-based settings.

PDAs may be useful in a place such as Malaysia, particularly for appropriate SDM in diabetic patients. Type 2 diabetes has risen rapidly in Malaysia to 18.3% prevalence in 2019 [8] and is a significant factor for cardiovascular disease that is the leading cause of death in Malaysia [9]. As nearly three-fourths of Malaysian diabetic patients are unable to achieve glycemic targets [10], insulin is now recommended for early treatment [11]. However, there are a number of factors and misconceptions that make Malaysian patients reluctant to initiate insulin therapy such as fear of pain and injections, risks for kidney failure, and the perception that insulin therapy indicates end stage diabetes [12–15]. Hence, a PDA for insulin therapy in diabetes has been created in Malaysia, available in the local languages (Malay, Chinese, Tamil and English) to cater to the Malaysian population (https://decisionaid.ohri.ca/AZsumm.php?ID=1558) [16].

Although the insulin PDA has been piloted [17], formal implementation and integration for regular use in public health clinics in Malaysia have not been conducted. Therefore, this study aims to identify perceived factors influencing implementation of the insulin PDA in public health clinics in Malaysia. The findings of this study hope to inform future development of common and contextual implementation strategies to incorporate PDA use into standard clinical practice in public health clinics under the same organization.

## Method

### Study design and participants

This study adopted a comparative case study design with a qualitative focus [18, 19]. This study design was chosen because it allowed identification of similarities and differences of

potential barriers and facilitators to implementing the insulin PDA across different as well as within implementation sites.

In this study, in-depth interviews (IDIs) with clinic managers and focus group discussions (FGDs) with healthcare providers (HCPs) and patients were conducted. All components were conducted to ensure quality data was collected; clinic managers were not included in FGDs with HCPs for fear that this would change power dynamics in the group except for Clinic C where the clinic manager and the HCPs requested to be grouped together. FGDs with HCPs were separate from patients, so that FGDs would include participants that are peers to enable richer and more robust findings. However, for participants who were unable to participate in FGDs, IDIs were conducted to accommodate their schedules.

The participants of this study were purposively sampled and consisted of 1) clinic managers who have the authority to decide which health intervention should be implemented in the clinic, 2) HCPs (family medicine specialist, medical officer, diabetes educator, staff nurse, pharmacist) who are involved in advising patients about starting insulin and 3) patients with type 2 diabetes who have been seeking treatment in the clinics and advised to, or, are currently using insulin. In literature surrounding SDM and PDA implementation, patient's perspectives on implementation processes are often not reported even though they are the end-users of the innovation. Patient can contribute such as by informing from who and how they want the innovation to be delivered to them [20–22]. Patients are included in this study to obtain a more comprehensive finding on potential factors that could influence the insulin PDA implementation is obtained.

## Settings

This study was conducted in five public health clinics located in the area of Klang Valley and Selangor, Malaysia; an upper middle-income country with a multiethnic population comprised of three main ethnicities namely Malay (67.4%), Chinese (24.6%) and Indian (7.3%) [23]. These five clinics fall under the Malaysian Ministry of Health [24]. Diabetes patients in Malaysia are largely seen at these clinics where insulin initiation is mainly conducted due to the availability of the resources and the fact that insulin treatment are subsidized. In the public health clinics, implementation of innovations are often conducted using the top-down approach without taking into consideration the needs of the frontline staff who puts the intervention into practice. It should be noted that while the public health clinics are under the same organization, the clinics vary in terms of locality, leadership, patient population and work culture.

The researcher contacted 12 public health clinics by e-mail to inform them of the purpose of the research and they were then asked about the characteristics of their clinics before finally selected five clinics selected to seek participation. These clinics were selected based on their variation in terms of location, patient population profile (ethnicity, socio-economic status, education level) and presence of insulin support and manpower for diabetes management (Table 1).

## Study instrument

A semi-structured interview guide was utilized for this study. The development of the interview guides were informed by the Theoretical Domains Framework (TDF) [25], literature review and discussions among researchers. The TDF was selected because implementation or adoption of the insulin PDA is primarily dependent on behavior change of the HCPs and patients. The TDF consists of 14 theoretical domains synthesized from 33 behaviour change theories and can guide in development of interventions targeting at behavior change. Besides

**Table 1. Characteristics of the public health clinics selected.**

| Characteristic | Clinic A | Clinic B | Clinic C | Clinic D | Clinic E |
|---|---|---|---|---|---|
| **Location** | • Urban | • Urban | • Urban | • Semi-urban | • Semi-urban |
| **Patient profile (Ethnicity*, socio-economic status, education level)** | • Predominantly Chinese and Indian<br>• Middle to high income group<br>• High education level | • Predominantly Chinese and Indian<br>• Middle to low income group<br>• High and low education level | • Predominantly Malay<br>• Middle income group<br>• High education level | • Predominantly Chinese<br>• Middle to low income group<br>• Low education level | • Predominantly Malay and Indian<br>• Low income group<br>• Low education level |
| **Presence of insulin support for patients in the clinic** | • Diabetes team present<br>• Diabetes clinic operates weekly morning<br>• DMTAC by pharmacists | • Diabetes team present<br>• Has a diabetes clinic<br>• DMTAC by pharmacists | • No diabetes team<br>• No diabetes clinic<br>• DMTAC by pharmacists | • Diabetes team present<br>• Diabetes clinic operates Thursday morning weekly<br>• DMTAC by pharmacists | • Diabetes team present<br>• Diabetes clinic operates weekly morning<br>• DMTAC by pharmacists |
| **Manpower (staffing) in diabetes management in the clinic** | • 2 MOs in-charge of diabetes<br>• One diabetes educator<br>• 2 staff nurses in diabetes clinic<br>• A dietician | • 6 MOs in-charge of diabetes<br>• One diabetic educator<br>• 2 staff nurses in diabetes clinic<br>• A visiting dietitian from the State Health Department will come twice a month.<br>• One medical assistant | • 4 MOs in-charge of diabetes (21 MOs in clinic are on rotation for diabetes clinic)<br>• One diabetes educator<br>• No staff nurses<br>• A visiting dietitian who goes to clinic once weekly. | • One family medicine specialist in-charge<br>• One MO in-charge<br>• One diabetes educator<br>• 2 nurses trained in diabetes management<br>• One dietician<br>• One optometrist | • Two MOs in-charge<br>• One diabetes educator<br>• One nurse performing the tasks of a diabetes educator<br>• A visiting dietitian from the State Health Department comes three times a week |

DMTAC: The Diabetes Medication and Therapeutic Adherence Counseling service; MO: Medical officer;

* In Malaysia, ethnicity is proxy for the types of language spoken in an individual. Malaysia has a multiethnic population, which comprised of three main ethnicities namely Malay, Chinese and Indian. The national language in Malaysia is the Malay language, however, while Chinese and Indians can speak in the Malay language there are also some who may not have high Malay language proficiency. Chinese and Indians also have their own language namely Mandarin and Tamil. English is generally widely spoken in Malaysia however the proficiency varies from high to low as English is generally considered as second language.

taking account of rational and cognitive process, the TDF also covers emotional and organizational factors such as working environment and resources [26] that can help to elicit barriers and facilitators to implementation or adoption of the insulin PDA. Questions and prompts were created according to the domains in the TDF and adapted according to healthcare policy-maker, HCPs and patient participants (S1 Appendix).

## Data collection

The researchers of this study are WTT, YKL, NCJ and PYL. WTT is a PhD student while YKL, CJN and PYL are lecturers. CJN and PYL are also clinicians who specialize in family medicine. All researchers are experienced in conducting qualitative research. The data collection was conducted from July to September 2016. WTT sent e-mail invitations to clinic managers to participate in the study. Prior to this research, YKL, CJN and PYL have collaborated with clinic manager C in the development of the insulin PDA thus clinic manager C was aware of the availability of the insulin PDA. Other clinic managers knew YKL, CJN and PYL given that they were all in the field of family medicine but only at the level of acquaintance. All clinic managers agreed to participate and site meetings were set to know more about the clinic context as well as provide more information to the clinic managers. During the site meetings, WTT and YKL were brought around the sites and explained on the clinics' running processes.

Through this, the researchers were able to observe and understand each of the clinic contexts better. Then, the clinic managers were subsequently asked to refer researchers to staff who fulfilled the participant inclusion criteria for this study. In majority of the clinics, all the HCPs in the diabetes team were referred for participation, as they were the most relevant individuals except for Clinic C, which was without a diabetes team. The clinic manager of Clinic C selected the HCPs to participate. Appointments were then made using telephone with individuals who agree to participate to conduct the interviews. Patient participants were recruited face-to-face by seeking help from the medical officers who were practising in the clinic on the day the interviews were conducted with the clinic managers and the HCPs.

Participant response rate was 100% for both HCPs and patients. All the interviews were conducted in the clinics for the convenience of the participants. Only the interviewer and the interviewees were present during the IDIs while for FGDs, a note taker was also present to help capture verbatim notes. No repeat interviews were conducted. WTT conducted most of the interviews (13/19 IDIs; 7/9 FGDs). To ensure the study rigour, YKL, CJN and PYL were refrained from conducting interviews. However, YKL conducted a few IDIs and FGDs when some of the interviews had to be carried out concurrently. Being the developers of the insulin PDA, YKL, CJN and PYL were aware of their own biases and tried their best to distance their personal judgments when carrying out the data collection and analysis.

Prior to the interviews, the researchers gave the study information sheet and explained to the participants the purpose of the study, information on the insulin PDA, the concept of SDM and the various PDA modalities (booklet, tablet, website) available before informed consent was obtained from the participants. Given that SDM and PDA is a foreign concept in Malaysia, a video, which demonstrates how the insulin PDA can be used during a consultation, was also shown to all the participants to give them a clear picture on one of the ways the insulin PDA can be implemented in the clinic. The purpose of showing the video was to provide the participants with a clearer understanding of SDM and how the insulin PDA can be operationalize so they could think of the potential factors that would influence its implementation in their clinic settings. After viewing the video, the participants were asked to think about the possible barriers and facilitators and other ways the PDA can be implemented in the clinic. They were also assured that their responses would be kept confidential and their participation in this study would not affect their work. The interviews were conducted by WTT and YKL. All the interviews were audio-recorded and they lasted on average 50–90 minutes. Field notes were taken to capture data that cannot be audio-recorded such as observations on the clinic surrounding and participants' non-verbal gestures. Interviews ceased when there was no new information that emerged from the participants in each of the clinic (data saturation). In addition, interview reflections were also noted at the end of each interview sessions to capture researchers' views about the interview sessions as well as the points that has been raised by the participants.

## Data analysis

The interviews were transcribed verbatim and checked by the researcher (WTT) before imported into NVivo qualitative software for analysis. Thematic analysis was conducted. Initially, transcripts were read line by line and codes (short phrases label) were assigned to specific data sections that represented their significance (open coding). Then, codes that were developed were reviewed and group together to form categories (axial coding). Related categories were then reviewed and overarching themes were applied that reflected the meaning of the data. Once the coding framework was developed, it was then used to code data from other transcripts (selective coding) [27]. Table 2 shows the development of the coding frame.

**Table 2. Development of the coding frame.**

| Category | Theme |
|---|---|
| • Time constraint due to high patient and work load<br>• Lack of manpower | Time constraint |
| • Lack of funding (to print PDA booklets and purchase computer)<br>• Clinic lack of facilities to print PDA booklets | PDA costs |
| • Patients cannot read or understand the insulin PDA<br>• Language barrier between HCP and patient | Tailoring PDA use to patient profile |
| • Patients let doctors or relative to make health decisions for them<br>• Patients lack of confidence in using the insulin PDA by themselves to make decision | Patient decisional role |
| • Having a senior role model<br>• Acknowledgement on using the insulin PDA by clinic authority | Leadership and staff motivation |

The data analysis was performed case by case for each clinic at a time in order to gain in-depth understanding on the contextual factors that influence implementation of the insulin PDA at the specific setting. Initially, the researchers WTT and YKL read the transcripts in detailed from Clinic A and coding was performed independently. Categories and themes emerged were discussed and finalized when discrepancies were resolved. The finalized themes and categories for Clinic A were later used as a coding framework for data analysis for the rest of the clinics. Any new codes and categories that emerged were added to the coding framework while those that were not relevant were removed. WTT coded all the transcripts for all the clinics while YK coded for Clinic B and C, PYL coded for Clinic C and clinic D and CJN coded for Clinic E, independently. Align with the comparative case approach, which used various forms of data [19], field notes from the sites observations and interview reflections were also referred to help in understanding and interpretation of the data. Furthermore, data analysis were also performed sequentially starting from Clinic A to Clinic E that allowed WTT to fully immerse into the data for each clinic as well as making comparisons to other clinics as categories and themes emerged [19]. Subsequently, the themes and categories were compared across the five clinics and the similarities and differences of the findings were identified. The findings comparison across the clinics was indeed challenging, time consuming and exhausting as the researchers had to go through the transcripts, field notes, interview reflections for both HCPs and patients data from each clinics back and forth in order to understand the data thoroughly. However, by going back to the data again and again, soon the patterns of commonalities and differences in the barriers and facilitators across the clinics became clear to the researchers.

## Ethics considerations

This study received ethics approval from the Medical Research and Ethics Committee Ministry of Health Malaysia (NMRR-15-1598-27260) and the University of Malaya Medical Centre Medical Ethics Committee (reference: MECID.NO: 20158–1600).

## Results

A total of 19 IDIs and 9 FGDs were conducted with a range of 43 stakeholders (policymaker: 5; doctor: 9; pharmacist: 6; diabetes educator: 3; pharmacist: 6; patients: 15) from the five clinics. Table 3 shows participants' socio-demographic information by clinic. Detail information of individual participant in each clinic can be found in S2 Appendix.

**Table 3. Participants' socio-demographic information by clinic.**

| | Clinic A | | Clinic B | | Clinic C | | Clinic D | | Clinic E | |
|---|---|---|---|---|---|---|---|---|---|---|
| | HCP (n = 5) | Patient (n = 3) | HCP (n = 5) | Patient (n = 2) | HCP (n = 5) | Patient (n = 4) | HCP (n = 8) | Patient (n = 4) | HCP (n = 5) | Patient (n = 2) |
| **IDI** | 1 | 3 | 5 | 0 | 0 | 1 | 3 | 2 | 2 | 2 |
| **FGD** | 2 | 0 | 0 | 1 | 2 | 1 | 1 | 1 | 1 | 0 |
| **Age (years)** | 36 ± 7.81 | 69 ± 7.5 | 37 ± 8.9 | 39 ± 4.2 | 35.2 ± 7.8 | 61.8 ± 2.9 | 39.8 ± 9.7 | 54.3 ± 5.5 | 35.2 ± 10.6 | 45.5 ± 12.0 |
| Mean ± SD (Range) | (29–49) | (61–76) | (30–49) | (36–42) | (29–48) | (58–65) | (29–59) | (47–59) | (29–54) | (37–54) |
| **HCP duration of practice in the clinic (years)** | | | | | | | | | | |
| Mean ± SD | 3.5 ± 1.6 | n.a | 3.7 ± 2.1 | n.a | 3 ± 4 | n.a | 5.9 ± 4.7 | n.a | 4.5 ± 3.15 | n.a |
| (Range) | (1–5) | | (7 months–6) | | (11 months–10) | | (1–13) | | (1–9) | |
| **Patient duration of seeking treatment at the study clinic (years)** | | | | | | | | | | |
| Mean ± SD | n.a | 5.3 ± 1.1 | n.a | 8.0 ± 2.1 | n.a | 3.5 ± 2.6 | n.a | 14.2 ± 6.1 | n.a | 5.5 ± 2.1 |
| (Range) | | (4–6) | | (5–11) | | (2 months—6) | | (8–20) | | () |
| **Ethnicity** | | | | | | | | | | |
| Malay | 2 | 0 | 3 | 2 | 4 | 4 | 5 | 3 | 4 | 2 |
| Chinese | 1 | 2 | 1 | 0 | 1 | 0 | 1 | 0 | 0 | 0 |
| Indian | 2 | 1 | 1 | 0 | 0 | 0 | 1 | 1 | 1 | 0 |
| **Highest education level** | | | | | | | | | | |
| Primary | 0 | 0 | 0 | 0 | 0 | 0 | 0 | 3 | 0 | 0 |
| Secondary | 0 | 3 | 0 | 1 | 0 | 3 | 0 | 1 | 0 | 1 |
| Undergraduate | 2 | 0 | 2 | 1 | 3 | 1 | 3 | 0 | 4 | 1 |
| Postgraduate | 3 | 0 | 3 | 0 | 2 | 0 | 5 | 0 | 1 | 0 |
| **Position** | | | | | | | | | | |
| Clinic manager | 1 | n.a | 1 | n.a | 1 | n.a | 1 | n.a | 1 | n.a |
| Medical officer | 2 | n.a | 2 | n.a | 2 | n.a | 2 | n.a | 1 | n.a |
| Diabetes educator | 1 | n.a | 1 | n.a | 1 | n.a | 1 | n.a | 0 | n.a |
| Staff nurse | 0 | n.a | 0 | n.a | 0 | n.a | 3 | n.a | 1 | n.a |
| Pharmacist | 1 | n.a | 1 | n.a | 1 | n.a | 1 | n.a | 2 | n.a |
| **Patient using insulin** | | | | | | | | | | |
| Yes | n.a | 2 | n.a | 2 | n.a | 2 | n.a | 4 | n.a | 2 |
| No | n.a | 1 | n.a | 0 | n.a | 2 | n.a | 0 | n.a | 0 |

Data analysis uncovered five themes related to perceived factors that could influence the implementation of the insulin PDA in the five clinics and they were: 1) time constraint, 2) PDA costs, 3) tailoring PDA use to patient profile, 4) patient decisional role, and 5) leadership and staff motivation. Based on the interviews and drawing on observations and interview reflection notes, the researchers identified that theme 1–2 emerged as common prominent factors that cut across all the clinics, however themes 3–5 were more prominent for certain clinics.

## Theme 1: Time constraint

Time constraint was raised as a major potential barrier for implementation of PDA for all five clinics. Even though the participants had positive views about the insulin PDA and were keen to use it, the large amount of patients attending the clinic may render them unable to go through the insulin PDA with patients in a thorough manner or to even use it at all.

*"This clinic has one of the highest load of patient so when we talk about using this PDA, definitely it's good but we have to take time in explaining to patients, reading. The time part is really constraining".*

*- FGD 1_Clinic A_MO 2*

*"I don't think our doctors have the time to explain to patients. The amount of patients is almost 1000 patients, do you think doctor will open the book (PDA)?"*

*- FGD 3_ Clinic B_Patient 1*

All the clinics felt that while diabetes educator and nurse could take up the role of delivering the insulin PDA to patients in order to address the time constraint faced by doctors, however, they were reported to be performing other tasks unrelated to their role due to the lack of manpower in the clinic. Only one diabetes educator is available in some clinics and this was felt to be inadequate. Staff nurses were seen as unsuitable to deliver the PDA to patients as they are on a rotation basis, i.e. subjected to transfer from one division in the clinic to another.

*"I think if we have more diabetic educator and nurses they can also help to assist the doctors in providing the PDA by giving patients the PDA while they are waiting to see the doctor and talk about insulin initiation. Here, we only have one diabetic educator and we have trained 2 staff nurses to take the duty of diabetic educator, but the problem is our nurses are doing various works because of the lack of manpower".*

*- IDI 5_ Clinic B_Clinic manager*

## Theme 2: PDA costs

Printing of the insulin PDA booklet was another concern due to the lack of funds in the clinics. The clinics' financial constraints was highlighted in various scenarios such as the cease of subsidization of certain drugs for patients, delay in repair of clinic's infrastructure, discontinuation of effective programme and difficulty in purchasing papers for office use.

*"We don't have money to print. Our budget has been cut down 20%. Our funding comes from the state health office. Our priority is on drugs but even now we have been cutting down on some non-essentials drugs that we are not giving to patients anymore. Money is an issue".*

*- IDI 5_ Clinic B_Clinic manager*

*"This clinic has a lack of facilities to print. The government budget for the clinic decrease every year. It will be hard to get the clinic to print the PDA themselves."*

*- FGD 3_ Clinic B_Patient 1*

## Theme 3: Tailoring PDA use to patient profile

Among the clinics from semi-urban areas where many patients were reported to have limited education and lower socio-economic status (Clinic D and E), the clinic managers were concerned if the PDA would be applicable to patients in their setting due to patients' ability to comprehend the information in the insulin PDA. This was felt could be a prominent potential barrier for the insulin PDA implementation.

*"I have great doubts about how this book can be implemented in this clinic, whether at all. Literacy is very low here. I don't know whether the patients can even articulate what their concerns are. I think they might not even know how to use the PDA themselves".*

*- IDI 16_Clinic E_Clinic manager*

*"Some patients are not very well educated. So they may not understand what they read and what this is it all about."*

*IDI 7_ Clinic A_Patient*

Furthermore, majority of the patients in Clinic D were Chinese-speaking while Clinic E were Tamil-speaking. Most of the patients were not fluent or literate in Malay or English language (the common mediums used by HCP to converse with patients). Hence, this would make PDA implementation challenging, as effective communication is required to delivery SDM. While language barrier was also raised as a potential barrier for other clinics such as Clinic A and B, however, majority of their patients were educated and able to read. Most of their patients could speak either English or Malay fluently and these clinics also had a multi-ethnic staff team who could help with translation.

*"We have a lot of Indian patients so some of them might not be able to read the PDA in the Malay language. Even if we have a Tamil version, I don't think the doctors here can read it if the patients write it (in Tamil)".*

*- IDI 16_Clinic E_Clinic manager*

*"Most of them (patients) can speak in Malay so I don't think is a problem".*

*- IDI 2_ Clinic A_Patient*

*"The patients in this clinic are from upper class. Many of them are Dato, Tan Sri (honorary titles confer to an individual by Malaysian state ruler. The titles are quivalent to the British 'Sir'). There are some patients who do not have high education level who might need someone to explain to them but this group of patients is small here. Many patients are well educated and can read."*

*- FGD 3_Clinic B_Patient 1*

## Theme 4: Patient decisional role

Patients desire for involvement in SDM with doctors vary between clinics where there were more patients who were educated and from the middle-income group (Clinic A and B) compared to clinics where many patients had limited education (Clinic D and E). It was noted that majority of the patients in the latter clinics rely on their doctors and family members to make health decisions for them, especially the older patients, hence the PDA may not be suitable to be used with such patients.

*"Shared decision making is mainly letting the patient to make the decision themselves, but our patients of the older generation rely on doctor to decide for them. So far, I've not come across any who said 'Ok doctor I will do this, I will think over it and decide'".*

*- IDI 11_Clinic D_Clinic manager*

At clinics where patients were generally of high education level (Clinic A and B), patients appeared more empowered and would be more likely to be involved in decision-making. The credibility of the PDA would be an important criterion that the patients would consider as it was noted that some patients would actually look at the rank of the doctors when seeking consultations. In addition, patients may not be willing to return to the clinic to meet the nurses as compared to doctors should the insulin PDA is implemented.

*"Is important that we know it (the PDA) is from the universiti and not from any other places. Institution is very important."*

*–IDI 3_ Clinic A_Patient 2*

*"The patients here would only listen to the doctor. They don't listen to nurses. Later when they come back for appointment, they will say 'Oh it is only to see the nurses, never mind then'. That's the problem with educated patients laugh)".*

*- IDI 8_Clinic B_Diabetes educator*

## Theme 5: Leadership and staff motivation

Another potential factor influencing implementation of PDA is the clinic's leadership and staff motivation. Staff motivation appeared to be stronger in most of the clinics where specific time was allocated to diabetes team to attend to diabetes patients (Clinic A, D and E). Clinic A and D noted that the implementation of the insulin PDA can be carried without the need for directives from higher authority once they are educated on the advantages of the intervention. Staff training, empowerment and acknowlededgement will motivate the staff to implement the PDA at their own will.

*"I think they would be willing to do (implement the PDA), our diabetic team is very dedicated [laugh]. They will want to try something new".*

*- IDI 13_Clinic D_Family medicine specialist _*

*"The leader implementing it should train and acknowledge the staff for doing it. It means it is not just my project. It is a team work. I think acknowledging what the staff is doing is very important".*

*- IDI 12_Clinic D_MO1*

Nevertheless, having a diabetes team was not always a guarantee of PDA implementation as Clinic B demonstrated that leadership support was necessary. Drawing on observation and interview reflection notes, the current leadership appeared to be lacking in Clinic B and this might affect the insulin PDA implementation even though a diabetes team is present. Diabetes may not be Clinic B's health priority as the clinic manager's key-focus is on another health area. The Clinic B's manager also voiced that effective implementation of the insulin PDA would depend on the diabetes MO in-charge of diabetes in the clinic. However, during the researcher's (WTT) interview with the diabetes MO in-charge, the individual demonstrated a lack of interest with the insulin PDA and was also in a hurry to end the interview session. The HCPs in Clinic B have also reported to not like to spend time talking to patients and the insulin PDA may be used as a substitute for consultation. Furthermore, staff nurses may not have a sense of responsibility of using the insulin PDA as they are on rotation and may perceived that it is the diabetes educators' responsibility. At Clinic C, which was without a diabetes team, the

clinic manager noted that diabetes leadership is low in the clinic and this might lead to ineffective implementation of the insulin PDA. The diabetes educator was reported to have little enthusiasm while the doctor in-charge of diabetes is relatively new in the diabetes team having just been trained for one month (at the time of the interview) to coordinate diabetes management in the clinic. Furthermore, HCPs were also reported to try to get their work done in a rush due to patients' who are in a hurry to leave the clinic.

*"The problem here is we are rushing to finish seeing patients as soon as possible. I received a complaint because I was slow. The culture here is that if you see a patient and if there is no problem you just give medicine and they go home. Patients also complained (that the work process) here is slow. Here you need to finish seeing one patient in five minutes (laughs). Since the patients are like that, the staff also had to follow the flow. Patients will scold if is taking too long. That's why those who can read and understand and, has the patience they can take it home. For patients who are impatient, I don't think they will appreciate this one (insulin PDA)".*

*- FGD 5_Clinic C_MO2*

## Discussions

This study uncovered perceived factors influencing implementation of PDA in public health clinics in Malaysia. The findings highlighted that besides looking into macro level of implementation such as cost and provision of resources, healthcare authorities should take consideration of the micro contextual factors that affect implementation such as a clinic's patient population, leadership and staff motivation at specific clinic settings. There is a need for implementation strategies to be tailored to address the contextual needs at each implementation site as a one-size-fits-all approach may not be effective for implementation [28].

The findings of this study pointed to the unique implementation challenges that were prominent to some clinics in terms of their patient population such as patients' ability to comprehend the information in the PDA due to low literacy, language barrier and those who tend rely on others to make health decisions for them. When implementing PDAs, healthcare authorities need to be aware of the barriers faced by individual clinics in relation to their context and specific strategies have to be employed. The use of video images in PDAs, having HCPs to deliver PDAs to patients in a straightforward, plain language, tailoring the level of information to individual patients' needs and in the context of clinical care may help to facilitate patients' understanding of the information in the PDAs among those with low literacy [21, 29–31]. To address communication language barrier between HCP and patient during PDA discussions, presence of multi-ethnic staff in a clinic was noted to be solution in this study and healthcare authority can consider providing staff of different ethnicities to clinics, which faced the same barrier. To address the barrier of patients relying on HCPs to make health decisions, efforts to increase patients' awareness on being more involved in their health care and self-efficacy in making health decision, as well as participation in SDM are warranted. One study have shown that an intervention to reduce patients' unvoiced needs by getting patients to note down issues to be discussed with doctors prior to their clinical encounter to be effective [32] and may be adopted to faciliate PDA implementation.

This study also found that there is a variation in terms of leadership and staff motivation for diabetes promotion among the clinics. Getting support from clinic leadership and clinical champion is one of the key facilitators for effective PDA and SDM implementation [2, 33, 34]. To increase staff motivation to use the PDA, there is a need to make HCPs to recognize and

believe the benefits of practicing SDM and use of PDAs. At the healthcare system level, incentivising the use of PDA may be the key to motivating HCPs to be involved in the implementation. One example is such as making PDA use as an alternative means of informed consent to protect HCPs from 'failure to inform' lawsuit [35]. At the broader level of PDA implementation is the need to inculcate SDM culture and practice and HCPs need to be taught on the concept of SDM and PDA. HCPs need to understand their role in SDM in terms of providing quality information to patients and supporting patients in their deliberation of the treatment option [36]. SDM is generally a foreign concept in Malaysia in that SDM activities only began in 2010 [37]. While the Ministry of Health Malaysia and the Malaysian Medical Council has encourage doctors to perform SDM-related activities such as providing patients with accurate and timely information, discuss treatment limitation and risks, and collaborate with patient in selection of treatment since 1990s [37–39], patient involvement in healthcare decision making in Malaysia was still found to be lacking as there is no formal training and education, laws, regulations and health policies that supports SDM [40]. SDM is not taught explicitly in the medical undergraduate and postgraduate curriculum [37]. To implement SDM and PDA effectively, HCP training is warranted. Studies have reported the lack of training to be contributing factors to lack of engagement of patients in SDM and proper use of PDAs [41, 42]. The benefits of training are well evidenced [43, 44] and it should be a pre-requisite for SDM and PDA implementation [45].

This study found that provision of resources such as manpower and funding by the Ministry of Health is not adequate for the clinics' current operation. This can have significant impact on PDA implementation or even implementation of other health innovations in general. It was found that the diabetes educators provided in the clinics had to perform other duties, which were not relevant to their job scope. Such would be a waste and inefficient use of the diabetes educator's expertise to help in the delivery of the PDA to patients. Involving diabetes educator in the implementation of PDA can help to avoid dependence on doctors in using the PDA with patients during consultation. Given that manpower resources provided by the Ministry of Health is already stretched, increasing manpower is not likely to be possible. Strategies similar to increasing existing staff motivation to be involved in the PDA implementation as described above can be adopted. In terms of the issue of funding, efforts are needed to obtain buy-in from the Ministry of Health to invest in PDA implementation so that specific allocation of the national healthcare funding can be provided. There is a need to show healthcare authority on the advantages of PDA in terms of improved patient care [1] as well reduced healthcare costs [46]. More local evidences are needed on the positive impact of the insulin PDA implementation to clinic services as well as patient health outcomes to obtain the clinics' higher authorities buy-in to prioritise the insulin PDA implementation. An alternative to address the cost of PDA implementation is the use of web-based PDAs, which preclude the need for printing costs and can be access freely by patients.

## Study limitations

This study has a few limitations. Social desirability bias may have been present with participants who were favorable to either the use of the PDAs or those that wanted to provide a favorable view of their clinic. In addition, findings from the data analysis were not provided to the participants for feedback. However, the researcher ensured that the data were accurately interpreted by crosschecking the analysis of the findings to interview reflections and memos that were jotted down at the end of each interview. Lastly, the few patients recruited as compared to HCPs in each clinic might have rendered patients' voice to be underrepresented. Nevertheless,

this study provided insights into patients' perspectives that were scarcely reported in literature surrounding PDA implementation.

## Conclusions

This study found time constraint as a major potential barrier for PDA implementation and effective implementation of the insulin PDA across different public health clinics would depend on leadership, staff motivation and the need to tailor PDA use to patient profile. Policy makers should avoid a 'one size fits all' approach when implementing PDAs in their health care settings.

## Supporting information

**S1 Appendix. Interview guide for healthcare policy maker, healthcare providers and patients.**
(PDF)

**S2 Appendix. Detailed information of individual participant in each clinic.**
(DOCX)

**S1 Checklist. Consolidated criteria for reporting qualitative studies (COREQ): 32-item checklist.**
(DOCX)

## Acknowledgments

The authors wish to thank the MOH Malaysia for permission to conduct and publish the study as well as the clinics involved in this study for the facilities and assistance provided.

## Author Contributions

**Conceptualization:** Wen Ting Tong, Yew Kong Lee, Chirk Jenn Ng, Ping Yein Lee.

**Data curation:** Wen Ting Tong, Yew Kong Lee, Chirk Jenn Ng, Ping Yein Lee.

**Formal analysis:** Wen Ting Tong, Yew Kong Lee, Chirk Jenn Ng, Ping Yein Lee.

**Funding acquisition:** Wen Ting Tong, Yew Kong Lee, Chirk Jenn Ng, Ping Yein Lee.

**Investigation:** Wen Ting Tong, Yew Kong Lee, Chirk Jenn Ng, Ping Yein Lee.

**Methodology:** Wen Ting Tong, Yew Kong Lee, Chirk Jenn Ng, Ping Yein Lee.

**Project administration:** Wen Ting Tong, Yew Kong Lee.

**Supervision:** Yew Kong Lee, Chirk Jenn Ng.

**Writing – original draft:** Wen Ting Tong.

**Writing – review & editing:** Wen Ting Tong, Yew Kong Lee, Chirk Jenn Ng, Ping Yein Lee.

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
