## [Decision Letter · Decision Letter 0]

2 Aug 2020

PONE-D-20-02666

Factors influencing implementation of an insulin patient decision aid at public community clinics in Malaysia: A qualitative study

PLOS ONE

Dear Dr. Lee,

Thank you for submitting your manuscript to PLOS ONE. I am extremely sorry for the delay in getting a decision to you but we had great difficulties finding reviewers for your paper.  After careful consideration, we feel that it has merit but does not fully meet PLOS ONE’s publication criteria as it currently stands. Therefore, we invite you to submit a revised version of the manuscript that addresses the points raised during the review process.

We look forward to receiving your revised manuscript.

Kind regards,

Chaisiri Angkurawaranon

Academic Editor

PLOS ONE

Journal Requirements:

2. Please address the following:

- Please include additional information regarding the interview guides used in the study and ensure that you have provided sufficient details that others could replicate the analyses. For instance, if you developed a guide as part of this study and it is not under a copyright more restrictive than CC-BY, please include a copy, in both the original language and English, as Supporting Information.

- Please ensure you have thoroughly discussed any potential limitations of this study within the Discussion section.

Reviewers' comments:

Reviewer's Responses to Questions

**Comments to the Author**

1. Is the manuscript technically sound, and do the data support the conclusions?

Reviewer #1: Yes

Reviewer #2: No

2. Has the statistical analysis been performed appropriately and rigorously? 

Reviewer #1: N/A

Reviewer #2: No

3. Have the authors made all data underlying the findings in their manuscript fully available?

Reviewer #1: Yes

Reviewer #2: No

4. Is the manuscript presented in an intelligible fashion and written in standard English?

Reviewer #1: Yes

Reviewer #2: Yes

5. Review Comments to the Author

Reviewer #1: General

This is an important and relevant topic. Implementation of patient decision aids in routine health care settings is indeed challenging. We do not have much reports from 'standard clinical practice', as the authors refer to the routine health care setting. Let alone from the Asia Pacific region.

The authors have succeeded to complete a labor-intensive task, in implementation and in qualitative data collection. The interview guide has been informed by a well-known theoretical framework.

Major revisions

- The real challenge, as implementation of Patient Decision Aids is not a goal in itself, is the implementation of a true dialogue and sharing of decision making attributes, values and preferences between doctor and patient. The authors could reflect on this overarching aim in the discussion.

- Methods: it is not clear to me why the authors used both in-depth interviews and focus groups? And, how they did decide on what to use when. It seems that focus groups were sometimes done with only 2 participants?

- Methods: Sampling a heterogeneous sample of participants is often a difficult task in qualitative research. Apparently, the contact person per clinic sampled the professional and patient interviewees. It may have the risk of bias, e.g. in selecting the less critical critical persons?

- Methods: It is not clear how the data collection was timed with respect to the implementation activities. Were the participants just in the beginning of the implementation period? Looking back after active implementation has ended? In the middle of the innovative change? (The first quote in theme 5 confused me. Is this a statement based on real experience. Or on expectation?

- I was a bit shocked by the quote in theme 2. If a budget cut of 20% (!) is being set, implementation of Patient Decision Aids seems impossible…. Should this barrier be given more weight in the paper?

- Theme 4, the 1st quote. “If SDM is looked upon as mainly letting the patient to make decision themselves…..” This is in my view a frequent occurring misconception of what SDM actually is about. It is NOT about the patient deciding for him or herself. Patients want support, want to share the decisional stress. While the doctor stays, of course, end-responsible for the decision. This may need a reflection in the discussion. Did the clinicians understand what SDM is really about?

Minor revisions.

- Table 1: the reporting of the estimates of crude overall numbers of patient populations per clinic would help in interpreting the differences between clinics.

- Did the authors gain any insight in the actual behavior with regard to the success of dissemination of PDAs? If so, this could be added to table 1.

- From table 2 it appears that mostly higher educated patients have been interviewed. Did this influence the findings?

- The evidence underlying the statement on saving national healthcare expenditure is actually really thin. I would grade down to “this may help….” Instead of “this helps”

- Discussion: interesting statement on involving family members in the implementation of patient decision aids. But how? And how about privacy issues?

Minor comments (Typos etc)

Introduction

- clarify -> clarifying

Methods

- Allow -> allowed

Results

- Avoid acronyms like DE

- In theme 3 one quote is printed twice

- Theme 4 share decision making -> shared decision making

- …listen to doctor -> listen to the doctor

- Oh is only -> Oh it is only

- Complain here is slow -> ???

Discussion

- Avoid acronyms like MOH

- Staff nurses has -> staff nurses have

- Doctors’ -> doctors

- They wanted the -> they wanted

- To patient -> to patients

- More involve -> more involved

- The intervention has been -> the interventions have been

- Involving og family -> involving family

- Support clinical champion -> support from a clinical champion

- Did increased -> did increase

Reviewer #2: Thank you for the invitation to review this manuscript.

This paper is a qualitative case series/study of implementation of patient decision aids (PDA) at community-level clinics in the Malaysia public health system. It aims to identify factors related to implementation of a PDA intervention for insulin treatment of T2DM. It looks at a series of community clinics and analyzes them as cases to draw comparisons and illustrate factors and barriers affecting implementation of a proposed PDA intervention.

Regrettably, I don’t think this manuscript is acceptable for publication at this stage. The quality of the methods, analysis and interpretation of the findings are lacking. Considerable work will need to be done to ensure that the methods are clear, and I recommend a conceptual framework (the Theoretical Domains Framework suggested by the authors in the Methods) to guide the analysis. As it is, the analysis is quite basic and simplified, leaving a lot to be desired depending on the audience, which we may presume to be policy makers or public health managers within the Malaysia Ministry of Health.

More specific comments are as follows. Please note this article has no numbered lines or page numbers (next time please consider when you submit), so I apologize for the difficulty in finding specific reference within the manuscript.

Abstract

1. “PDA” is not fully spelled out at its first mention.

2. It is not clear how the first sentence of the abstract relates to the second sentence.

3. “A comparative case study design with a qualitative focus” should be clarified better. More on this in the Methods sections.

4. Results: We start to get a sense that actually the analysis is basic, or it is summarized a bit simplistically.

5. The numbered list is a bit awkward, it is ok to list them without numbering the themes.

6. “HCPs” is not spelled out fully. Please review abbreviations, make sure they can be used in a PLOS ONE abstract, and I recommend using any abbreviations sparingly.

7. “Successful implementation”: what is successful? What is implementation? After reading the article, I am not always clear on how these terms are used in this manuscript.

Introduction

8. A fuller introduction would include discussion that addresses: How big of a diabetes burden is there in Malaysia? What is the proportion of those with diabetes who need insulin? Has the PDA been piloted elsewhere? How, where, and was it beneficial? If it is shown to work elsewhere, but maybe not everywhere (such as at a community level), then the purpose of your research will be stronger.

9. Also, there is no definition or clarification of what implementation means, what "successful" implementation looks like. Does it just mean implementing the PDA and utilizing it, or does it mean that it is utilized AND benefits patients? These are slightly different research questions, so "implementation" should be clarified.

10. Somewhere in the intro or in the methods, it is important to outline for the reader what the PDA is, who delivers it to the patients, and how. Because sometimes it sounds as if DE give the information, then it sounds like it's only the MO or doctors--it is unclear. If you are able to show how the different staff members in the clinics represented in table 1 participate (or don't participate) in PDA for insulin, that would greatly contextualize your findings and make it easier for your reader.

11. (http://dmit.um.edu.my/?modul=DMIT_PDA): it is unclear if this is the link that is meant. There is limited information available through this link.

12. “This study aims to explore the factors…”: I still fail to see why patients have been included at all as study participants.

Methods

1. IDI, FGD: these are not fully described. Which method was used with which type of participant? Why were these methods chosen for the specific participants, context, and research question? When you revise your methods, please follow the COREQ checklist: Allison Tong, Peter Sainsbury, Jonathan Craig, Consolidated criteria for reporting qualitative research (COREQ): a 32-item checklist for interviews and focus groups, International Journal for Quality in Health Care, Volume 19, Issue 6, December 2007, Pages 349–357, https://doi.org/10.1093/intqhc/mzm042

2. Table 1, “Patient profile (Ethnicity)”: Are the authors assuming that ethnicity determines language capacity? It seems from the results that language is more important then perhaps cultural elements that can be attributed to ethnicity. Would this be more important than ethnicity?

3. Table 1 “illiterate”: Are these participants illiterate in their own languages? Illiterate in Malay? And how is this determined?

4. Table 1 “Manpower”: One may wonder if manpower also relates to number of patients seen at a clinic. Should this also be mentioned as a defining characteristic of the different clinics?

5. Throughout the methods there is mention of FGD, then there is seemingly a description of only interviews in this section. Then we come to the results tables and we find that indeed, focus groups were performed. Who were included in interviews and who were included in focus groups and how were the focus group guides created?

6. It is unclear why showing the video on appropriate PDA use would have been part of the design? This also may bias the responses because you showed a way that PDA could be used?

7. Because it's not clear with whom IDI or FGD were conducted, these methods are not clear. Did you show a video to all participants, including diabetic patients?

8. Data analysis: The tools used for the interview guides are made to sound very strong. However, it seems that their potential for a deeper analysis was limited. The analysis could have been better guided by using the theoretical domains mentioned in the methods as a conceptual framework to perform your thematic analysis. In addition, if you use a case studies approach to your qualitative analysis, I recommend reviewing qualitative analysis texts: Creswell JW. Qualitative Inquiry & Research Design: Choosing Among Five Approaches. London: SAGE; 2007. Some good instruction on performing qualitative analysis: Miles MB, Huberman AM, Saldaña J. Qualitative data analysis: a methods sourcebook. 4 ed. Los Angeles: Sage; 2020. See Chs 4, 6, 9.

Results

9. Table 2: I don't find this table helpful. It needs to be summarized better by interview type and study site. You can then add a range of values for different characteristics--for example for the FGD.

10. This table also makes me wonder more about the methods: what determined which participants would be used in interviews or focus groups? It is clear that you had interviews with patients. But sometimes you mixed members of teams in a focus group at a given clinic? Would there be different bias inherent in the data collected from a discussion with professionals with different training, expertise, experience, hierarchy, power dynamics? Would this benefit/hurt the quality of the data collected? How? Why not conduct FGD with patients? It seems that this may be a more efficient way to elicit quality data from a patient perspective, and also enhance your ability to reach far more patients. These types of issues should be considered more fully in your Methods section. Again, refer to the COREQ checklist recommended above.

11. Paragraph after Table 2 “implementation”: I get the sense that this is not the appropriate term or technically what is being studied here. What does implementation mean? Since this is central to your study, it should be defined and discussed much earlier in your manuscript.

12. FGD 3_Clinic B_Patient 1 quote, “I don’t think our doctors don’t have time to explain”: Mistranslated? I believe it's meant that the doctors DON'T have time to explain.

13. Theme 2, IDI5_Healthcare policymaker_Clinic B, “We dno’t have money to print”: Implicit here is rationing and the need to prioritize at a local community level. However, it would be more interesting to hear why PDAs were not prioritized—and how might this contrast to other forms of patient education? I think if it is low priority it may give a deeper sense of why implementation may not be widespread or differ from clinic to clinic.

14. Theme 3, Patient profile: This section should be re-thought. If this is about SDM, this strikes me that the main theme emerging here is not the fault of the patients, but the fault of the PDA not being tailored to specific patient needs. I note that no patient accounts of how they are limited in understanding the materials, if that is what is being asserted, such evidence from patient interviews would support this. But even if the patients are limited, the crux of the problem lies in the materials being inappropriate, not particularly a problem with the patients themselves. If we follow the authors’ logic, one would conclude that the patient needs to be educated in the national languages or have to rise (or be raised) in their SES to get over their limitations to make PDA implementation possible. This can’t be what the authors intend to present and it can’t be what health policy makers would benefit from learning. A more thoughtful analysis would instead consider that the materials not being tailored to the particularly diverse Malaysian population is really the problem.

15. “The patient profile in the clinics pose distinct barriers”: I struggle that in a study trying to improve SDM, we still harp on the "patients" being portrayed as something of an obstacle. This reads a bit paternalistic, condescending.

16. IDI 16_Healthcare policy maker_Clinic E: I find two concepts in this quote, but I find one to be particularly problematic. The first part casts doubt on the suitability of the materials, but the second part has a bit of a condescending view towards the patients themselves. Perhaps if more policy makers felt this way, this becomes a barrier to implementation--the prejudice of those in management or higher positions that believe some inherent limitations of the patients are what prevent the PDA from being used. This strikes me as a prejudice among higher level officials that is problematic, less so than the patients' themselves. However, it could be bolstered by a patient perspective that cannot understand the material, but again, the theme here would be the appropriateness of the materials for PDA as they relate to whether or not they are utilized at a given clinic.

17. Theme 4, “The credibility of the PDA”: I point out comments above that this is a good example where a clear explanation of how and by whom the PDA information is given to the patient would be helpful to the reader. The PDA as you have put forth is provided by the doctor. So I am unclear what the nurses have to do with this?

18. Theme 5, “The emphasis on diabetes management in the clinic claimed to be moderate”: What does moderate mean?

19. Overall, this is a very basic qualitative analysis. The way it reads is a bit unclear what exactly the emergent themes were, and I am unsure about their validity in supporting the authors’ conclusions. It seems that the results focus on specific, mundane particulars about the clinics themselves without clear illustration of something more comprehensive picture that a strong thematic analysis would provide.

Discussion

20. “There is a need for implementation strategies to be tailored to address the contextual needs at each implementation site”: Yes, but unfortunately, I think your analysis was very limited if this was the reason for your study. You would have been better placed to take a bottom-up approach to your study design if this statement reflects the authors’ purpose. But instead, your study design took a top down approach, which the authors themselves argue is not appropriate in the introduction. Your data emphasizes policy makers, medical officers, other health staff, and lastly the patients.

21. “Efforts are needed to obtain buy-in from the ministry to invest in PDA…”: Yes, and this is why I recommend that in the introduction you state the burden of this particular problem. If you want policy makers to utilize your paper, cost of care information should also be provided if there is already existing data within Malaysia.

22. “There is a need to show healthcare authority…”: PDAs may be important, in general, but you haven't shown how they're important for diabetes (the literature you cite seldom discusses PDA forT2DM, important in the Malaysian context, of specifically how PDAs are useful to support insulin treatment in diabetes. If this is the assertion you want to make in the discussion, I would place the argument for it in the introduction.

23. “…as they were less empowered and rely on others…”: This is an overreach of your results. This was what providers said, not the patients. I think the results don’t do enough to illustrate the patient’s voice in this study.

6. PLOS authors have the option to publish the peer review history of their article (what does this mean?). If published, this will include your full peer review and any attached files.

Reviewer #1: **Yes: **trudy van der weijden

Reviewer #2: No

---

## [Author Response · Author response to Decision Letter 0]

27 Oct 2020

Please refer to the 'Response to reviewers' file attached.

---

## [Decision Letter · Decision Letter 1]

18 Nov 2020

PONE-D-20-02666R1

Factors influencing implementation of an insulin patient decision aid at public community clinics in Malaysia: A qualitative study

PLOS ONE

Dear Dr. Lee,

Thank you for submitting your manuscript to PLOS ONE. After careful consideration, we feel that it has merit but does not fully meet PLOS ONE’s publication criteria as it currently stands. Therefore, we invite you to submit a revised version of the manuscript that addresses the points raised during the review process.

We look forward to receiving your revised manuscript.

Kind regards,

Chaisiri Angkurawaranon

Academic Editor

PLOS ONE

Additional Editor Comments (if provided):

Thank you for the revisions. The reviewers still has some specific concerns which will likely help improve the clarity of your work. Please attend to them as best you can.

Reviewers' comments:

Reviewer's Responses to Questions

**Comments to the Author**

1. If the authors have adequately addressed your comments raised in a previous round of review and you feel that this manuscript is now acceptable for publication, you may indicate that here to bypass the “Comments to the Author” section, enter your conflict of interest statement in the “Confidential to Editor” section, and submit your "Accept" recommendation.

Reviewer #1: All comments have been addressed

Reviewer #2: (No Response)

2. Is the manuscript technically sound, and do the data support the conclusions?

Reviewer #1: Yes

Reviewer #2: Partly

3. Has the statistical analysis been performed appropriately and rigorously? 

Reviewer #1: N/A

Reviewer #2: Yes

4. Have the authors made all data underlying the findings in their manuscript fully available?

Reviewer #1: Yes

Reviewer #2: Yes

5. Is the manuscript presented in an intelligible fashion and written in standard English?

Reviewer #1: Yes

Reviewer #2: Yes

6. Review Comments to the Author

Reviewer #1: I have now understood that his paper is about PERCEIVED factors for effective implementation, which were explored among the stakeholders BEFORE actual implementation. Although a qualitative analysis of actual experience with a patient decision aid during or just after implementation in practice would in my view have been stronger, I still see value of the findings of this exploration of perceived barriers and facilitators. For reasons of clarity I suggest to always talk about perceived factors for implementation, at least in the abstract.

Overall, I'm in the opinion that the authors have improved the paper sufficiently along my comments.

Reviewer #2: Thank you again for the opportunity to review this revision. This is a revised manuscript of a qualitative study looking at factors in implementing a patient decision aid at community clinics in Malaysia.

It is clear that the authors have worked very hard on these revisions. Many of the original questions have been clarified, in terms of methods and results, but unfortunately, I feel this needs to be heavily revised, especially the Discussion which is very unfocused, restates or includes information that is more appropriate in the results, and is very hard to follow.

I hope the authors do not get too frustrated; I understand that in making sure that the manuscript had sufficient, pertinent information for clarity, now it has increased in length. Many of my suggestions on this round are to tighten the text, especially in the introduction, methods, and discussion.

Although this might be slightly different for qualitative studies, a good rule of thumb for dividing up a manuscripts is: introduction 10-15%, methods 15-20%, results 40%, discussion 30%. I have made some suggestions to get the introduction and methods a bit more balanced, but the results are still limited, and I feel this may be because some of the results are actually in the discussion—which is too lengthy.

Introduction:

Thank you for the effort to add pertinent information and make your case clearer. In responding to the reviewers, the length of the introduction has greatly increased, and there may be some revisions we can make to shorten it.

1) Is it possible to shorten the first three paragraphs into one paragraph? It seems we can get to the purpose of your paper more quickly: PDAs are a means to enhance patient-centered care, for SDM between patient and provider. Although shown to be effective in improving decision making, implementing PDAs in routine clinical settings can be hampered by xx, xx, and xx. Furthermore, there is limited data on implementing PDAs broadly in Asia. Then you can delete the third paragraph.

2) Thank you for comprehensively answering the questions about diabetes burden in Malaysia and the need for insulin therapy. Some of this may save some space in the introduction if it’s included in the Methods. For example, Lines 25-27 are more about the ‘setting”. I would omit this and consider including it in the “setting” section of the methods. Instead, begin this paragraph with a clear topic sentence: “PDAs may be useful in a place such as Malaysia, particularly for appropriate SDM in diabetic patients.”

3) Shorten the next three sentences: “Type 2 diabetes has risen rapidly in Malaysia to 18.3% prevalence in 2019, and is a significant factor for cardiovascular disease that is the leading cause of death in Malaysia.”

4) Next: “As nearly three-fourths of Malaysian diabetic patients are unable to achieve glycemic targets (REF), insulin is now recommended for early treatment (REF). However, there are a number of factors and misconceptions that make Malaysian patients reluctant to initiate insulin therapy such as fear of pain and injections, risks for kidney failure, and the perception that insulin therapy indicates end stage diabetes (ref).”

5) The next paragraph, some of the information can be cut and added to the Methods. I would focus the introduction to say, “Hence, a PDA for insulin therapy in diabetes has been created in Malaysia (25), available in the local languages to cater to the multiethnic, multilingual Malaysian population.”

6) Line 53-56 “The insulin PDA met…” can be saved for the methods.

7) Shorten Line 56-61 to: “Although the insulin PDA has been piloted, studies on implementation of the PDA and integration for regular use in community clinics in Malaysia have not been conducted.”

8) Then I would jump directly to the final paragraph: “Therefore, this study aims to identify factors influencing…”

9) I am a little confused with the mention of “prospectively” in line 73; from the study it seems like the review was done retrospectively after the PDA had been introduced in clinics? I note the comment to the other reviewer to clarify this, but to me, it seems a prospective study of this type would be a planned evaluation AFTER the implementation of a PDA, but the study design does not seem that it is so structured.

Methods:

10) Lines 81-86, the first three sentences can be shortened to one.

11) Lines 88-93 are quite repetitive; can you shorten to maybe 1-2 sentences?

12) Thank you for clarifying in-depth interviews, but for the wider audience, “individual” is not needed, I know this was included to help me; it is now clear because I know who will be interviewed (clinic managers). But here is some suggestion to state the most important information succinctly: “In-depth interviews (IDI) with clinic managers and focus group discussions (FGD) with HCPs and patients were conducted. All components were conducted to ensure quality data was collected; clinic managers were not included in FGD with HCPs for fear that this would change power dynamics in the group. FGD with HCPs were separate from patients, so that FGDs would include participants that are peers to enable richer and more robust findings.”

13) Please note, sometimes plural forms of the acronyms are used, e.g., “IDIs”, other times not; just a note to make sure the authors are consistent.

14) I would try to include lines 62-71 in the “Settings” section of the Methods instead of the introduction. I would take this paragraph and include it as a second paragraph in your “settings” section: “These five public health clinics fall under the Malaysia Ministry of Health. Diabetes patients in Malaysia are largely seen…work culture.”

15) I actually would take what has been written under “study participants” and add it to the end of the “Study design” section, and retitle the study design section, “Study design and participants.” Then it becomes much clearer who is included in the IDI vs FGD. Also, be careful to say that they are policy maker OR clinic managers, but using too many categories may be confusing for the reader. In the results you use mainly the word “health policymaker.”

16) “Study instrument” section is quite clear, well done.

17) Table 2, and the Results: the themes can be shortened. Can there be one or two words for “Time, patient load, lack of manpower?” “Cost to print PDA” can be just “PDA costs”? The other themes are appropriate I feel.

Results:

18) Table 3 is tricky. I find tables in qualitative data are very tricky. Theoretically, the case can be argued that static indicators, such as age, HCP duration, patient durations, are not needed to be so strictly summarized, as these are not going to be “representative” in a quantitative sense. As a reviewer, I feel IDI and FGD at the different clinic is helpful, but I would also like to see the breakdown of FGD for HCPs vs patients. The way the columns and rows are set up, the ethnicity, education level or exactly what participant is unclear. Again, as a reviewer, it would be nice to see the ethnicity and education level for FGDs, HCPs and patients separately. In the end, this may be easier to present as text instead of a table.

19) Lines 241-242 now a bit unclear because I originally requested to take away the numbers. Sorry!

20) Overall, the results are now reported at much greater depth. Well done.

Discussion

21) This is the section that may need some significant revisions. The length of this section actually suggests to me that although the Results are stronger, the Discussion includes information that likely should be in the Results.

22) The first paragraph, from lines 399 until 409, is very strong. After this point the discussion gets very wordy and confused. Aim to have a 5-paragraph discussion. Outline the main points to convey to the reader. Be very strict on each paragraph. Make sure that one of these paragraphs is the limitations. Anything that is special and important in the discussion that does not relate to implication or how this study fits with other literature, should be placed back in the results. Avoid restating the results in the Discussion.

23) The paragraphs currently go back and forth between implications of this study for PDAs in general, and implications from this study about PDAs for insulin implemented in Malaysia. Then there is discussion about what to do for enhancing SDM, as well. This makes for a discussion that is quite hard to follow.

7. PLOS authors have the option to publish the peer review history of their article (what does this mean?). If published, this will include your full peer review and any attached files.

Reviewer #1: No

Reviewer #2: No

---

## [Author Response · Author response to Decision Letter 1]

30 Nov 2020

Please refer to the "Response to the reviewers" file

---

## [Decision Letter · Decision Letter 2]

9 Dec 2020

PONE-D-20-02666R2

Factors influencing implementation of an insulin patient decision aid at public community clinics in Malaysia: A qualitative study

PLOS ONE

Dear Dr. Lee,

Thank you for submitting your manuscript to PLOS ONE. After careful consideration, we feel that it has merit but does not fully meet PLOS ONE’s publication criteria as it currently stands. The review has some remaining suggestions and minor comments.  Therefore, we invite you to submit a revised version of the manuscript that addresses the points raised during the review process.

We look forward to receiving your revised manuscript.

Kind regards,

Chaisiri Angkurawaranon

Academic Editor

PLOS ONE

Reviewers' comments:

Reviewer's Responses to Questions

**Comments to the Author**

1. If the authors have adequately addressed your comments raised in a previous round of review and you feel that this manuscript is now acceptable for publication, you may indicate that here to bypass the “Comments to the Author” section, enter your conflict of interest statement in the “Confidential to Editor” section, and submit your "Accept" recommendation.

Reviewer #2: (No Response)

2. Is the manuscript technically sound, and do the data support the conclusions?

Reviewer #2: Yes

3. Has the statistical analysis been performed appropriately and rigorously? 

Reviewer #2: Yes

4. Have the authors made all data underlying the findings in their manuscript fully available?

Reviewer #2: (No Response)

5. Is the manuscript presented in an intelligible fashion and written in standard English?

Reviewer #2: Yes

6. Review Comments to the Author

Reviewer #2: This is a qualitative evaluation of the implementation of patient decision aids for insulin treatment in Malay patients with diabetes.

Once again the authors have done a great job responding to this round of revisions.

Only minor comments remain and I recommend this manuscript being reviewed by a native English speaker. Try to change passive to active voice.

I leave it to the editor but I feel the Tables 3 and the supplementary materials may be excessive.

Abstract:

1. Line 8: PDA previously defined, so does not need to spell it out here.

2. Line 14: omit “implementation”

3. Line 15: “public community health clinics” but I prefer what is said in the methods, “public health clinics”, which would be consistent with the methods, too. Also need to make sure “public health clinics” is used consistently throughout the paper.

4. Line 20: can number these themes again as they are in the results.

5. Line 23: omit “the themes of”

6. Line 24: change “contrasted” to “varied”

7. Line 26: should read “limited education and lower socio-economic status”—I recommend using this phrase in the other sections as well.

8. Change “could” to “to” in lines 28 and 30.

9. Conclusions are a bit odd; in the results the authors state that the most important one is time constraints but this isn’t reflected in the conclusions? As mentioned below, need to make sure Conclusions in the Discussion match the Introduction.

Introduction:

10. Line 25 pg 3, “Although the insulin PDA” can be a new paragraph.

11. I hope the authors’ are happy with these revisions—I find this intro to now be clear, succinct, and strong.

Methods:

12. HCPs is defined in line 25 but can be defined when it is first mentioned earlier?

13. Table 1: I don’t think the columns need to be divided into multiple rows for each subheading. E.g., I think it’s find to have “Patient profile” where for clinic A there is no border between predominantly Chinese and Indian, middle to high income group, high education level. Same for the following subheadings and across the columns.

14. Lines 230-231: not needed to include the degrees, but the second sentence in line 231 is sufficient.

Results: No comments

Discussion

15. This can be shortened. The main bias is “social desirability” reflected by the manager interviewed as well as the mangers’ selection of HCPs to participate. But I don’t think this level of detail is necessary. “Social desirability bias may have been present with participants who were favorable to either the use of the PDAs or those that wanted to provide a favorable view of their clinic.” This and the fifth limitation can be kept and grouped together as they are both answered by the last sentence. The fourth limitation is a good one, but add that you included patients where most of the other literature doesn’t look at this important participant group. I would omit limitations two and three.

16. Make sure the conclusions in the Discussion are the same as those in the abstract.

7. PLOS authors have the option to publish the peer review history of their article (what does this mean?). If published, this will include your full peer review and any attached files.

Reviewer #2: No

---

## [Author Response · Author response to Decision Letter 2]

11 Dec 2020

Please refer to "Response to reviewers" file

---

## [Editor Report · Decision Letter 3]

15 Dec 2020

Factors influencing implementation of an insulin patient decision aid at public community clinics in Malaysia: A qualitative study

PONE-D-20-02666R3

Dear Dr. Lee,

We’re pleased to inform you that your manuscript has been judged scientifically suitable for publication and will be formally accepted for publication once it meets all outstanding technical requirements.

Kind regards,

Chaisiri Angkurawaranon

Academic Editor

PLOS ONE

Additional Editor Comments (optional):

Thank you very much for the revisions.
---

## [Editor Report · Acceptance letter]

17 Dec 2020

PONE-D-20-02666R3 

Factors influencing implementation of an insulin patient decision aid at public health clinics in Malaysia: A qualitative study 

Dear Dr. Lee:

I'm pleased to inform you that your manuscript has been deemed suitable for publication in PLOS ONE. Congratulations! Your manuscript is now with our production department. 

Kind regards, 

on behalf of

Dr. Chaisiri Angkurawaranon 

Academic Editor

PLOS ONE